# Enhanced X-ray Emissions Arising from High Pulse Repetition Frequency Ultrashort Pulse Laser Materials Processing

**DOI:** 10.3390/ma15082748

**Published:** 2022-04-08

**Authors:** Jörg Schille, Sebastian Kraft, Dany Kattan, Udo Löschner

**Affiliations:** 1Laserinstitut Hochschule Mittweida, University of Applied Sciences Mittweida, Technikumplatz 17, 09648 Mittweida, Germany; kraft@hs-mittweida.de (S.K.); loeschne@hs-mittweida.de (U.L.); 2Institut für Laser-und Plasmaphysik, Heinrich-Heine-Universität Düsseldorf, Universitätsstraße 1, 40225 Düsseldorf, Germany; dany.kattan@hhu.de

**Keywords:** X-ray, ultrashort pulse, laser, plasma, dose rate, Bremsstrahlung, resonance absorption

## Abstract

The ongoing trend in the development of powerful ultrashort pulse lasers has attracted increasing attention for this technology to be applied in large-scale surface engineering and modern microfabrication. However, the emission of undesired X-ray photon radiation was recently reported even for industrially relevant laser irradiation regimes, causing serious health risks for laser operators. In the meantime, more than twenty influencing factors have been identified with substantial effects on X-ray photon emission released by ultrashort pulse laser processes. The presented study on enhanced X-ray emission arising from high pulse repetition frequency ultrashort pulse laser processing provides new insights into the interrelation of the highest-contributing parameters. It is verified by the example of AISI 304 substrates that X-ray photon emission can considerably exceed the legal dose rate limit when ultrashort laser pulses with peak intensities below 1 × 10^13^ W/cm² irradiate at a 0.5 MHz pulse repetition frequency. The peak intensity threshold value for X-ray emissions decreases with larger laser spot sizes and longer pulse durations. Another key finding of this study is that the suction flow conditions in the laser processing area can affect the released X-ray emission dose rate. The presented results support the development of effective X-ray protection strategies for safe and risk-free ultrashort pulse laser operation in industrial and academic research applications.

## 1. Introduction

Ultrashort pulse lasers (USPLs) have become established as a versatile tool for high-efficiency and high-precision micromachining, which is a result of their excellent laser beam performances. Moreover, the ongoing trend toward higher laser powers enables high throughputs and short processing times in modern microfabrication, and, thus, the USPL will shortly move forward from the laboratory state to industrial production. However, the emission of X-ray photon radiation constitutes a secondary laser beam hazard from USPLs in materials processing that can cause serious health risks for laser operators. The undesired X-ray photons can potentially arise when high-intense laser beams interact with solid surfaces, producing hot, dense plasma including very hot electrons of several keV. Since earlier studies in the 1970s, three main processes are known for X-ray radiation fields generated by ultrafast laser-excited electrons: (i) the Bremsstrahlung continuum arising from free-free transitions of accelerated free electrons, (ii) the recombination continuum from free-bound electron transitions, and, finally, (iii) characteristic line emissions originating from bound-bound transitions of the inner-shell electrons of ionized atoms. The energy distribution of the Bremsstrahlung X-ray photons follows a Maxwell–Boltzmann distribution where the released X-ray emission dose rate can be approximated by integration over the entire spectral X-ray photon flux. The X-ray emission dose released by a single laser pulse ranges typically from only a few pico Sievert (pSv) to several nano Sievert (nSv), and the potential risk for laser operators is fairly low. The X-ray emission dose rates and associated hazards and risks can considerably increase when employing high-average-power USPLs in materials processing. This is the result of the amazing progress in the development of modern USPL systems towards kilowatt average power that is delivered by ultrashort pulses at high pulse repetition frequencies (PRF) ranging between ten to hundreds of megahertz (MHz) [1]. In this case, millions of ultrashort pulses interact with the substrate surface in a short time, and the X-ray photons released by each single pulse accumulate to high X-ray dose levels, reaching up to 1 Sv/h [2,3]. In fact, such high X-ray emissions exceed innumerably the limit values for the general population and occupationally exposed persons in a calendar year, which are specified in the German Radiation Protection Act, and can cause serious health damage to the exposed parts of the human body.

So far, the onset of X-ray photon emissions has been detected for peak intensities above 10^13^ W/cm^2^, which can be delivered, for example, by ultrashort pulses of 1 ps pulse duration and a 10 J/cm^2^ fluence. This fluence is about ten times beyond the optimum fluence for the energy-efficient laser processing of metals [4,5,6] and can be achieved by tightly focusing the laser beam of µJ level optical pulse energy to spot sizes in the range of ten micrometers in the focal plane. In a recent article, the emission of harmful X-ray photon dose rates was demonstrated for much lower fluence or, rather, intensity, which was even below the legal peak intensity limit [7]. This was achieved under specific processing circumstances when MHz PRF pulses were irradiated at a small geometrical pulse distance.

A critical review of the actual literature in this field identified more than 20 factors affecting the X-ray photon emissions arising from ultrashort laser pulses irradiating solid surfaces (see Table 1). These factors also influence each other and can be subdivided into three different categories: (i) laser-sided, (ii) process-sided, and (iii) material-sided parameters of influence. First of all, on the laser side, the peak intensity of the laser pulse can be suggested as the most likely parameter, as doubling the peak intensity increases the released X-ray photon emissions about ten times. Therefore, when considering pulses of similar peak intensities, higher X-ray emission dose rates are detected for longer pulses and higher pulse energies. Another universal expectation from the results presented so far is that the X-ray emissions scale up almost linearly with increasing average laser powers. Furthermore, the PRF was figured out in a recent study on substantially affecting the released X-ray photon flux [7]. This was, in particular, the case when MHz PRF pulses irradiated with high spatial pulse overlapped at submicrometer spatial distances between the individual pulses within a laser-processed line. For the resulting strong increase in X-ray photon emissions, efficient plasma resonance absorption of the irradiating laser beam was suggested as the enhancing effect. The hypothesis is supported by the fact that MHz PRF pulses with very short time delay can interact with the still-remaining plasma plume induced by the preceding pulses [8]. So far, resonance absorption in the laser-produced plasma has been reported only for high-intense laser pulses where both the laser beam polarization and the angle of incidence have considerable influence on the total absorption of the incident laser beam [9,10,11]. Accordingly, it is validated in the presented article that, even for low-intense pulses, X-ray emissions can increase to high levels when the polarization of the MHz PRF laser beam is aligned parallel to the beam moving direction. This provides evidence that plasma resonance absorption might also be effective in low-intense and high PRF ultrashort pulse laser regimes. In addition, the suction flow speed in the processing zone is introduced as another influencing factor on X-ray emission dose rate. It is shown that reproducible X-ray emissions can only be detected in the case of controlled and stable suction flow conditions. A third key finding of this presented study is that X-ray photon emissions can be detected at low peak intensities up to five times below the legal peak intensity limit for approval-free laser operation in Germany. This can occur under certain processing conditions, e.g., when high PRF pulses of picosecond pulse durations irradiate at a small intra-line distance, the polarization of the laser beam is aligned parallel to the laser beam moving direction, and the suction flow speed is comparably low.

## 2. Experimental Section

Two types of USPL systems were utilized in this study: a UFFL 100 (Active Fiber Systems GmbH, Jena, Germany) and an FX 200 (Edgewave GmbH, Würselen, Germany). This provided a wide range of complementary laser beam parameter settings, i.e., 87 W maximum average laser power, up to 50 MHz PRF, and pulse durations from 340 fs ≤ *τ*_H_ ≤ 10 ps (sech² pulse shape). Near-infrared laser beams of 1030 nm wavelength were passed through individual beam expanders and were individually expanded and aligned through an f-theta focusing objective of 167 mm focal length. This resulted in a similar laser spot radius *w*_0_ in the focal plane, as determined by *w*_0/UFFL_ = *w*_0/FX200_ = 15.0 µm (1/e²-method), for the beam caustic measurements (Micro Spot Monitor, Primes GmbH, Pfungstadt, Germany). In addition, a galvanometer scan system (intelliScan 30, Scanlab GmbH, Puchheim, Germany) was assembled in the beam path for raster-scanning the laser beams in a line pattern across the studied stainless steel (AISI 304) substrate surfaces. The intra-line pulse distance *d*_X_, representing the geometrical distance of the impinging laser pulses within a scanned line, was varied as function of scan speed *v*_S_ and PRF *f*_P_ according to the following relationship: *v*_S_ = *d*_X_ × *f*_P_. The distance between the scanned lines in the line pattern was set by the hatch distance *d*_Y_. The polarization state of the linearly polarized laser beams was adjusted by a half-wave retarder plate aligned either parallel ($\vec{E}∥{\vec{v}}_{\mathrm{scan}})$ or perpendicular $\left(\vec{E}⊥{\vec{v}}_{\mathrm{scan}}\right)$ to the laser beam scan direction. In terms of safety precautions, all the experiments carried out in this study were performed inside a laser safety enclosure to protect the operators from direct and scattered laser beams, X-ray radiation, and other secondary harmful effects originating during high PRF ultrashort pulse laser processing.

The maximum pulse energy used in this study was *Q*_P_ = 36 µJ. The pulse energy was calculated from the measured average laser power divided by the PRF, taking into account the optical losses in the beam path. The peak power of a laser pulse *P*_0_ was calculated as the pulse energy divided by the pulse duration. Following Equation (1), the peak intensity of the laser pulses *I*_0_ was given by the pulse peak power divided by the focus spot area *A*_f_, considering a pulse-shape-dependent numerical factor of *f*_S_ = 0.88 for the sech² ultrashort pulses:(1)$$I_{0}=f_{S}\;\frac{P_{0}\;}{A_{f}}=f_{S}\;\frac{2\;Q_{P}\;}{\tau_{H}\;\pi \;w_{0}^{2}}$$

Accordingly, the maximum peak intensity of the ultrashort pulses was *I*_0/UFFL_ = 2.7 × 10^13^ W/cm^2^ and I_0/FX_ = 1.5 × 10^13^ W/cm^2^ for the UFFL 100 and FX 200 lasers, respectively. Notably, these peak intensity values were only slightly above an intensity of 1 × 10^13^ W/cm^2^, where the X-ray emission dose rate can exceed 1 µSv/h [3,7,12]. In fact, according to the German Radiation Protection Regulations, both the values 1 × 10^13^ W/cm^2^ and 1 µSv/h at a 100 mm distance from the accessible area represent the legal limits for approval-free laser operations [22].

The X-ray emissions arising from the laser beam with AISI 304 substrate interaction were analyzed using two different detectors: a survey meter OD-02 (STEP Sensortechnik und Elektronik Pockau GmbH, Pockau-Lengefeld, Germany) measuring the directional X-ray emission dose *H*′(0.07) and a SILIX lambda detector (Ingenieurbüro Prof. Dr.-Ing. Günter Dittmar, Aalen, Germany) for monitoring the X-ray emission spectra and X-ray emission dose rate ${\dot{H}}^{\prime }$(0.07) in the soft X-ray photon energy range between 2 and 20 keV. In each experiment, the detectors were aligned parallel to the laser beam scan direction towards the center of the laser-processed area. The dimensions of the raster-scanned processing field were 20 × 20 mm². The distance between the detectors and the scanning field center point was 200 mm, and the detection angle was 35°. A unidirectional scan regime was applied with a “laser-ON” cycle, with the laser beam moving away from the X-ray detector receiving maximum X-ray emission dose levels [13]. The laser ON/OFF duty cycle was determined to be 46% in oscilloscope measurements; thus, the monitored values were corrected by a factor of 2.17 to discuss process-typical X-ray emissions in the following sections. The background X-ray radiation in the laboratory was determined below 0.1 µSv/h, which was negligibly small compared to the detected laser-generated X-ray dose levels. Furthermore, it should be mentioned that in each case the X-ray emissions were captured from the first scan crossing of the laser beam over the substrates and were obtained after every six seconds. Respectively, the presented X-ray emission dose values represent mean values averaged from at least ten individual X-ray measurements. The standard deviation of the measured values is indicated in the diagrams.

## 3. Results

### 3.1. X-ray Emission as a Function of Intra-Line Pulse Distance

First, the X-ray emission dose per pulse *H*′(0.07)_PP_ was investigated as a function of the intra-line pulse distance and PRF. Therefore, ultrashort pulses with a peak intensity of *I*_0_ = 1.5 × 10^13^ W/cm^2^ were irradiated at *f*_P_ = 0.5 MHz and 1.6 MHz. The intra-line pulse distance was varied in the range of 0.15 µm < *d*_X_ < 2.0 µm by increasing the scan speed from 75 mm/s to 3200 mm/s, depending on the respective PRF, while the hatch distance was kept constant at *d*_Y_ = 20 µm. The X-ray emission dose per pulse, as shown in Figure 1, was derived from the averaged SILIX detector readings captured in individual test series on four different days. At *f*_P_ = 0.5 MHz, the highest X-ray emission dose per pulse of *H*′(0.07)_PP_ = 1 pSv was detected at 0.2 µm. For the larger intra-line pulse distances, there was a clear tendency towards lower X-ray emissions. Therefore, at *d*_X_ = 2 µm (representing the widest studied distance) the X-ray emission dose per pulse reduced up to two orders of magnitude to *H*′(0.07)_PP_ = 0.02 pSv, which emphasized the great effect of intra-line pulse distance on laser-induced X-ray emissions.

At higher PRFs (for example, 1.6 MHz in Figure 1 (right)), the highest X-ray emissions were found in the range of *d*_X_ = 1 µm, steadily decreasing with larger intra-line pulse distances. The highest X-ray emission dose per pulse was assessed at an *H*′(0.07)_PP_ = 16.0 pSv maximum with about 8.5 pSv average value. In comparison to low PRF pulses, this was more than a ten-fold increase referring to the maximum value and almost a hundred times higher than the X-ray emission dose arising at *d*_X_ = 1 µm and 0.5 MHz PRF. This strong increase in the X-ray emissions observed for the high PRF pulses provided evidence of stronger laser beam interaction with plasma during processing, thus, further enhancing X-ray photon generation, which will be discussed more in detail in the following subchapters. However, a significant drop in the X-ray emission dose rate per pulse occurred when 1.6 MHz pulses irradiated at small intra-line pulse distances (*d*_X_ < 1 µm), which was in contrast to the X-ray emission characteristics observed at 0.5 MHz. Therefore, a potential explanation can be found in the increased roughness of the high-PRF-processed surfaces, which could behave similar to the X-ray shielding recently observed on kerf walls and boreholes [3,11]. In the particular case of high PRF pulses irradiating at small intra-line pulse distances, self-organizing microscopic surface structures emerged at the substrate surfaces [23]. These rough and protruded surface features potentially screened the laser-induced X-ray photon flux, which in turn caused the lower monitored X-ray emission doses at small intra-line pulse distances seen in Figure 1 (right). In order to validate the proposed influencing effect of surface roughness on X-ray emission, X-ray photon fluxes arising from the laser processing of four different laser-pretextured substrates were analyzed, Figure 2.

For surface pretexturing, a laser beam of *I*_0/FX_ = 1.5 × 10^13^ W/cm^2^ and *f*_P_ = 1.6 MHz was raster-scanned across the AISI 304 substrates at a *d*_Y_ = 20 µm hatch distance. The intra-line pulse distances were varied between *d*_X_ = 3.0 µm (Figure 2a), 1.5 µm (Figure 2b), 0.9 µm (Figure 2c), and 0.6 µm (Figure 2d). In this way, specific surface roughness conditions were provided for the next impinging pulses similar to the ones discussed above in Figure 1. In the X-ray measurements, the laser processing conditions were as follows: *I*_0/FX_ = 1.5 × 10^13^ W/cm^2^, *f*_P_ = 1.6 MHz, *d*_X_ = 0.9 µm, and *d*_Y_ = 20 µm.

The highest X-ray emission dose per pulse of *H*′(0.07)_PP_ = 9.1 pSv was captured for the smooth surface, which was laser-pretextured with *d*_X_ = 3.0 µm and *S*_a_ = 0.49 µm corresponding area roughness, Figure 3. For the *d*_X_ = 1.5 µm laser-pretextured surface and resulting *S*_a_ = 0.94 µm, the X-ray emission dose per pulse was marginally lower in the range of *H*′(0.07)_PP_ = 8.2 pSv. These X-ray emission doses were almost equal to the ones measured for the nontextured basic material of similar surface roughness, *S*_a_ = 0.44 µm. For the rougher surfaces featured with area roughness measures of *S*_a_ = 1.8 µm and 6.8 µm resulting from laser pretexturing at smaller intra-line pulse distances of *d*_X_ = 0.9 µm and 0.6 µm, respectively, the detected X-ray emission doses per pulse amounted to *H*′(0.07)_PP_ = 5.4 pSv and 0.9 pSv, respectively. Remarkably, these dose values were even below the X-ray emissions arising from laser processing of the nontextured steel surface at *d*_X_ = 0.6 µm in Figure 1 (right), which was proof for the screening of laser-induced X-ray photons by rough surface features.

However, a closer look at Figure 1 shows a large variance of the X-ray emission dose levels, diverging up to a factor of ten for the measurements carried out on four consecutive days. An influencing effect of the laser beam or, rather, the ambient air conditions on the X-ray emission dose can largely be ruled out, as the laser beam parameters were carefully controlled during the study. Moreover, the experiments were performed in an air-conditioned laboratory room (temperature: 22.5 ± 0.5 °C; humidity: 44% ± 2%) in order to provide constant processing conditions.

### 3.2. Influencing Effect of Dust and Fume Extraction on X-ray Emission

After a detailed inspection of the experimental conditions, both the position of the fume extraction element (FEE) related to the processing area, as well as the suction flow speed *v*_fs_, were identified as potentially influencing factors for the monitored different X-ray emission dose levels. For a more precise investigation of this effect, the distance of the FEE from the processing area *d*_eff_ was varied during the measurements of X-ray photon emission arising from laser surface processing with pulses of *I*_0/FX_ = 1.5 × 10^13^ W/cm^2^, *f*_P_ = 1.6 MHz, and *d_X_* = 0.75 µm. As another control parameter, the suction flow speed was monitored at the center position of the laser-processed field using a resistance anemometer. The measurements showed a continuous reduction in the suction flow speed with increasing distance of the FFE from the processing area, Figure 4. At about a 190 mm distance, the suction flow turned from laminar to turbulent, which was indicated by the larger deviation of the measured suction flow speed. From this position, no further reduction in the suction flow speed was detected with larger distances because the cross jet air flow protecting the focusing objective was more predominant within the processing area.

The results obtained in this set of experiments validated a strong impact of the surrounding air flow on X-ray photon emission for the first time. In the laminar flow regime, the X-ray emission dose increased almost linearly with lower suction flow speed, while the maximum X-ray photon flux was measured in the turbulent flow regime. Hence, the experiments of Figure 1 were repeated under controlled suction flow conditions. As a result, the X-ray emissions arising from laser processing under stable suction flow speeds (*v*_fs_ = 0.9 m/s ± 0.3 m/s) showed a considerably reduced standard deviation ranging within the measurement accuracy of the SILIX detector. This was proven for both *f*_P_ = 0.5 MHz and 1.6 MHz repetitive pulses studied on two different days, Figure 5.

In Figure 5, the critical intra-line pulse distances *d*_X,crit_ are indicated. These values represent the geometrical pulse distances within a raster-scanned line where the maximum X-ray emission doses were detected. For the low PRF pulses, Figure 5 (left), the highest X-ray emission was achieved at *d*_X,crit_ = 0.15 µm, which was similar to the intra-line distance shown in Figure 1 for the highest X-ray dose level. For high PRF pulses, in contrast, the highest X-ray emission dose was recorded at *d*_X,crit_ = 0.75 µm, Figure 5 (right). This intra-line distance was achieved under stable suction flow conditions and was a little smaller than those for uncontrolled laser processing conditions, further underlining the influence of suction flow or, rather, the prevailing dust and fume concentration surrounding the laser processing area on X-ray photon emission.

### 3.3. X-ray Emission as a Function of PRF and Polarization State

A detailed analysis of the results obtained showed that the critical intra-line pulse distance for the maximum X-ray emission dose was strongly affected by the specifically applied PRF. For example, the critical intra-line distance was *d*_X,crit_ = 0.2 µm at *f*_P_ = 0.5 MHz, enlarging to *d*_X,crit_ = 0.9 µm at *f*_P_ = 1.4 MHz and above, Figure 6. Therefore, a potential explanation can be found in the complex interaction of the laser beam with the plasma plume induced by preceding pulses in a pulse train. Moreover, in the particular case of low-intense ultrashort pulses irradiating at MHz PRF, plasma resonance absorption might be the dominant absorption mechanism for the laser beam, which was already evidenced in a previous work [7]. This, in turn, implies optimum plasma characteristics in terms of the degree of ionization, plasma carrier density, dimension and flank angle of the critical plasma density layer, etc., for the most efficient coupling of impinging optical pulse energies with the laser-induced plasma plume. In fact, plasma plume formation and expansion are highly dynamic processes [8]. Therefore, it became evident for the dynamically changing plasma plume that both timing (defined by PRF) and position (defined by intra-line pulse distance) of the next impinging laser pulse had a great effect on optimum laser beam coupling resulting maximum X-ray photon emission. The general trend of increasing X-ray emission dose and enlarging critical intra-line pulse distance with higher PRFs can be seen in Figure 6 for two different suction flow speeds: *v*_fs_ = 0.9 m/s and 2.8 m/s. In addition to the X-ray emission dose per pulse, for a better comparison to the literature data, the measured values are also presented in the form of X-ray emission dose rate per irradiated laser power. The higher X-ray dose values and the larger critical intra-line pulse distances were found for the lower suction flow speed, which is in line with the data presented above in Figure 4. In this context, it is worth mentioning that the highest X-ray emission doses can be seen in Figure 6 with about same value of 0.8 mSv/h/W for PRFs ranging between 1.0 MHz and 1.8 MHz. This broad range of MHz PRF for the highest X-ray emissions seems to be contrary to previous results (e.g., Figure 5 (right) shows a distinct maximum at 1.6 MHz). The X-ray emission doses of Figure 6, however, represent maximum values that were achieved at different intra-line pulse distances.

In addition to PRF and intra-line pulse distance, the polarization state of the laser beam was another enhancing effect of ultrashort pulse laser-induced X-ray emissions, Figure 7. This applies in particular to MHz pulses and linear polarization aligned parallel to the beam scanning direction. When comparing with the perpendicularly polarized laser beam, a maximum four-fold increase in X-ray emission was figured out for the *f*_P_ = 1.0 MHz repetitive pulses irradiating at a *d*_X,crit_ = 0.5 µm critical intra-line pulse distance, Figure 7 (left, middle). At the higher PRF of 1.6 MHz, the X-ray emission of the parallel-polarized laser beam was about twice the X-ray dose obtained with perpendicular polarization, Figure 7 (left, bottom). A difference by the factor of two between the X-ray emission dose levels was also observed for the 0.5 MHz pulses irradiated either parallel or perpendicularly polarized with respect to the laser beam scanning direction, Figure 7 (left, top). The maximum X-ray emission dose levels of Figure 7, ranging up to *H*′(0.07)_PP,max_ = 1 pSv, were almost one order of magnitude below the values presented above in Figure 5 of *H*′(0.07)_PP,max_ = 10 pSv. This is mainly due to the fact that different suction flow speeds were applied (*v*_fs_ = 0.9 m/s in Figure 5 vs. 2.8 m/s in Figure 7), which further confirms the strong impact of suction flow speed on X-ray photon emission already shown above in Figure 6. From these results, it can be postulated that the suction flow speed manipulated the shape of the plasma plume, which, in turn, affected the efficient coupling of the following laser pulses into the plasma under resonant absorption conditions. The proposed influencing effect of laser beam polarization on X-ray emission can also be assessed from the X-ray emission spectra provided in Figure 7 (right) showing higher X-ray photon yields and wider X-ray spectra that were further enlarged to higher X-ray photon energies for the parallel-polarized laser beam.

Nevertheless, the strong dependence of X-ray emission spectra and dose rate on laser beam polarization is a further indication of plasma resonance absorption as an effective absorption mechanism when MHz PRF pulses interact with well-formed plasma states. This is validated by the fact that the polarization of laser beams has already been proven as an influencing factor for resonance absorption in high-intense laser-produced plasma. The originality of our study, however, is the evidenced resonance absorption of following pulses (irradiating with short time delays at MHz PRF) on the well-formed plasma plume induced by the preceding pulses in a pulse train. This was also confirmed by the monitored X-ray photon spectra and corresponding X-ray dose rates increasing disproportionately from ${\dot{H}}^{'}$(0.07) = 2.7 mSv/h to 9.0 mSv/h at 1.0 MHz or, rather, ${\dot{H}}^{'}$(0.07) = 7.0 mSv/h to 21.0 mSv/h at *f*_P_ = 1.6 MHz when using a parallel- instead of a perpendicular-polarized laser beam.

### 3.4. Influence of the Peak Intensity on X-ray Emission

In this study, maximum X-ray emission dose rates up to ${\dot{H}}^{'}$(0.07) = 35 mSv/h were produced with pulses of comparably low peak intensity somewhat above 10^13^ W/cm². As a rough estimate, the doubling of the peak intensity could induce 10 times higher X-ray emission dose rates, which implied that peak intensity is a significantly influencing variable of X-ray photon emission. Following Equation (1), the peak intensity denotes the maximum optical power related to the laser spot area and can be varied either by the pulse energy, the pulse duration, or the spot size of the irradiating laser beam. For a better understanding of the specific effect of these individual parameters on X-ray emission and further to identify a threshold value for the start of X-ray photon generation, the peak intensity was gradually reduced by (i) increasing the pulse duration between 330 fs ≤ *τ*_H_ ≤ 10 ps, (ii) reducing the pulse energy by lowering the average laser power to a constant *f*_P_ = 0.5 MHz, and (iii) increasing the laser spot area by placing the substrate surface out of the focal plane (defocusing). In the case of defocusing, the effective laser spot radius *w*_86_*(z)* of the laser beam depended on the distance of the substrate surface from the focal plane, hereinafter referenced as defocusing distance *z*. In order to define the peak intensity of the ultrashort pulses specifically applied in the defocused substrate position, according to Equation (1), the effective laser spot radius was calculated following Equation (2) by taking into account the Rayleigh length of *z*_R_ = 0.6 µm and the focus spot radius of *w*_0.86_ = 15.0 µm
(2)$$w_{86}\left(z\right)=w_{0.86}\sqrt{1+\frac{z^{2}}{z_{R}^{2}\;}}$$

In Figure 8 (left), the pulse energy showed the greatest effect on the released dose rate when comparing the X-ray emissions from pulses of varying peak intensities. This was validated by the steepest curve growth (blue, squares) in the ${\dot{H}}^{'}$(0.07) vs. *I*_0_ log-log plot. At a fixed peak intensity, the pulses of longer pulse duration induced higher X-ray emission dose levels (green, circles). The results also confirm that pulse duration affected the threshold intensity value, where X-ray photons were detected for the first time. For example, the intensity threshold was figured from *I*_0/UFFL_ = 1.3 × 10^13^ W/cm^2^ for 340 fs pulses. In contrast, with the longer 10 ps pulses, the X-ray photon emission started at *I*_0/UFFL_ = 2.3 × 10^12^ W/cm^2^. This was about four times below the legal peak intensity limit for approval-free laser operation in Germany. Moreover, the corresponding ${\dot{H}}^{'}$(0.07) = 10 µSv/h for the applied low-intense 30 W average power laser beam exceeded considerably the permitted 1 µSv/h maximum X-ray emission dose rate.

The spot size of the laser beam at the substrate surface was varied by defocusing. For larger spot sizes with focus positions above instead of inside the laser beam, the substrate surface showed up to three times higher X-ray emission dose rates. A potential explanation, therefore, can be found with the heel effect by screening X-ray photons generated inside the craters on the rough and protruded surface features. As a matter of fact, in the case of placing the focus position inside the material, the outer areas of the laser beam were cut off by the original surface, reducing the effective peak intensity of the laser beam travelling to the deeper areas being processed. The main effects of laser beam defocusing on X-ray emissions are indicated in Figure 8 (right), showing larger laser spot radii, lower peak intensity, and decreasing X-ray emissions for the greater distances of the laser focus plane from the substrate surface.

## 4. Summary and Main Conclusions

The emission of undesired X-ray photon radiation in ultrashort pulse laser materials processing poses a secondary laser beam hazard and can cause serious health risks for laser operators. The generation of harmful X-ray dose levels is affected by a wide range of laser parameters and processing conditions. In total, more than 20 influencing factors on the laser-, process- and material-sides have been identified as considerably affecting the X-ray emission dose, while the individual parameters also can have a direct influence on each other. In this article, in addition to peak intensity as the most likely variable, the PRF, intra-line pulse distance, polarization direction, and suction flow speed were shown as other contributing parameters enhancing laser-induced X-ray emissions. Based on the results obtained in the presented study, the following key findings were concluded:X-ray emission dose levels could increase up to ten-fold when ultrashort laser pulses irradiate at MHz PRF and submicrometer intra-line pulse distances;A critical intra-line distance depending on PRF existed for maximum X-ray emissions;Higher X-ray emissions could be released from smoother surfaces;The suction flow conditions in the laser processing area affected X-ray photon emissions, where higher suction flow speeds induced lower X-ray dose levels;A laser beam parallel-polarized to the beam moving direction induced higher X-ray emissions, which further confirmed plasma resonance absorption as an efficient laser beam absorption mechanism in a low-intense ultrashort pulse laser regime;Harmful X-ray emission dose rates exceeding the legal limit of 1 µSv/h could be generated with peak intensities below 1 × 10^13^ W/cm, and, finally;The peak intensity threshold value for X-ray emissions decreased with larger laser spot sizes and longer pulse durations.

These summarizing statements provide new insights into the fundamental mechanisms for X-ray photon emissions arising from USPLs applied in materials processing. Essentially, the detailed understanding of the interrelation between the main contributing parameters is particularly helpful for the development of effective X-ray protection strategies for safe and risk-free operation of powerful USPLs in industrial and academic research applications. Furthermore, the presented X-ray emission dose rates monitored in industrial-relevant USPL processes can provide a basis for the correct selection and dimensioning of X-ray radiation shielding enclosures.

## Figures and Tables

**Figure 1 materials-15-02748-f001:**
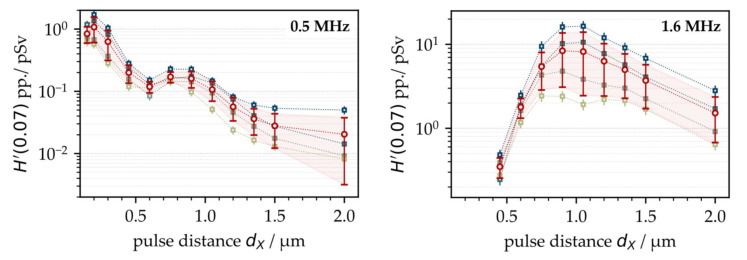
X-ray emission dose per pulse as a function of the intra-line pulse distance at 0.5 MHz (**left**) and 1.6 MHz (**right**) PRF obtained on 4 different days. The mean values are highlighted by the red symbols (circles).

**Figure 2 materials-15-02748-f002:**
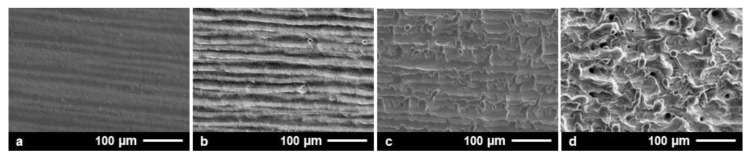
Laser surface textures produced on AISI 304 by raster-scanning of the laser beam across the substrates at 3.0 µm (**a**), 1.5 µm (**b**), 0.9 µm (**c**), and 0.6 µm (**d**) intra-line pulse distance.

**Figure 3 materials-15-02748-f003:**
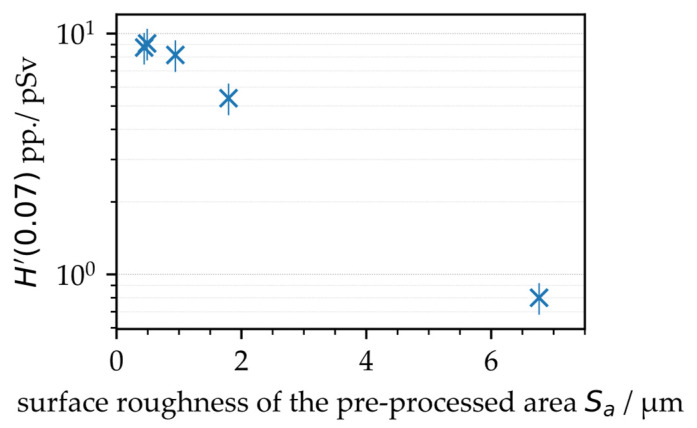
X-ray emission dose per pulse as a function of the area roughness of specifically laser pre-textured AISI 304 substrate surfaces.

**Figure 4 materials-15-02748-f004:**
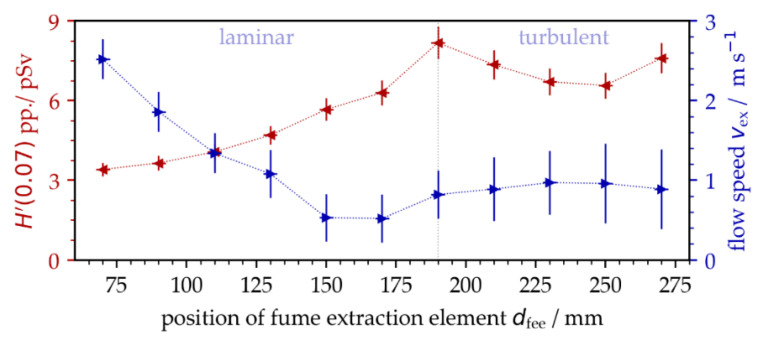
X-ray emission dose per pulse as a function of suction flow speed that was varied by increasing the distance of the fume extraction element from the laser processing area.

**Figure 5 materials-15-02748-f005:**
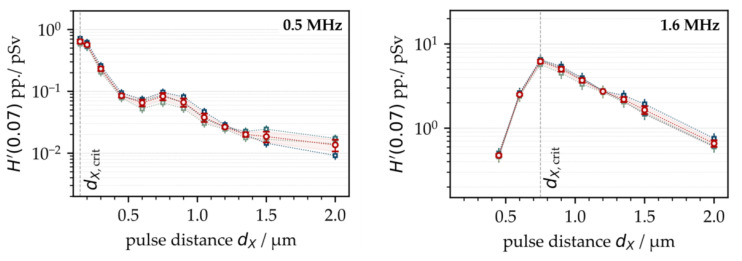
X-ray emission dose per pulse as a function of the intra-line pulse distance at 0.5 MHz (**left**) and 1.6 MHz (**right**) PRF obtained on 2 different days under controlled suction flow conditions. The suction flow speed was kept constant at 0.9 m/s ± 0.3 m/s. The mean values are highlighted by the red symbols (circles). The critical intra-line pulse distances d_X,crit_ are indicated.

**Figure 6 materials-15-02748-f006:**
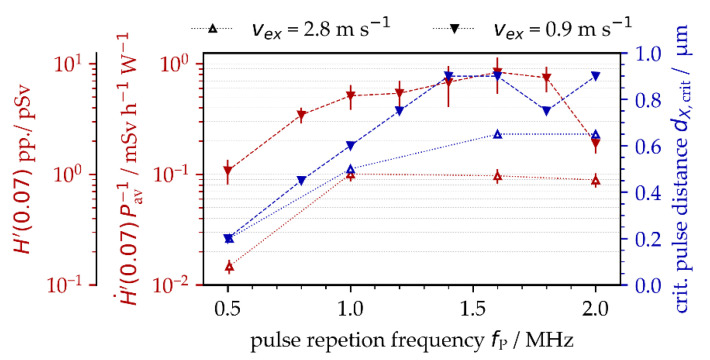
X-ray emission doses per pulse and X-ray emission dose rates related to the applied average laser power as a function of PRF detected at different suction flow speeds. The critical intra-line pulse distances for maximum X-ray emissions are presented.

**Figure 7 materials-15-02748-f007:**
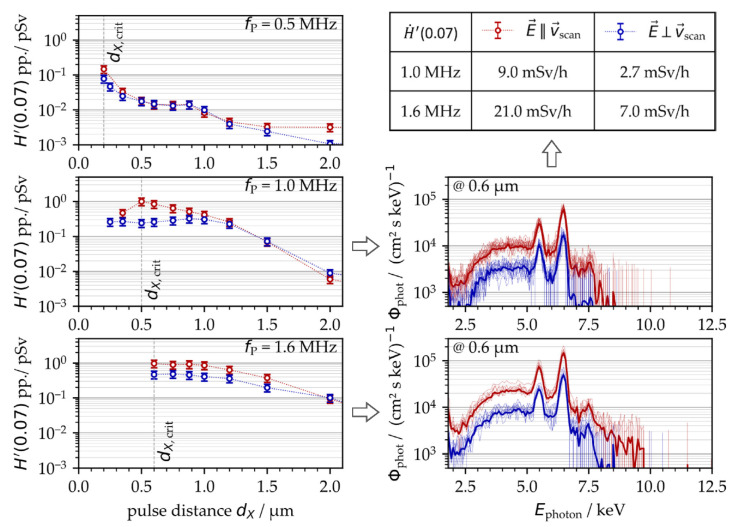
(**Left**): X-ray emission dose per pulse as a function of pulse distance, PRF, and polarization direction at a 2.8 m/s ± 0.4 m/s suction flow speed; (**right**): corresponding X-ray photon spectra monitored for 1.0 MHz or 1.6 MHz PRF, 0.6 intra-line pulse distance, 0.9 m/s ± 0.3 m/s suction flow speed, and either parallel (red) or perpendicular (blue) laser beam polarization.

**Figure 8 materials-15-02748-f008:**
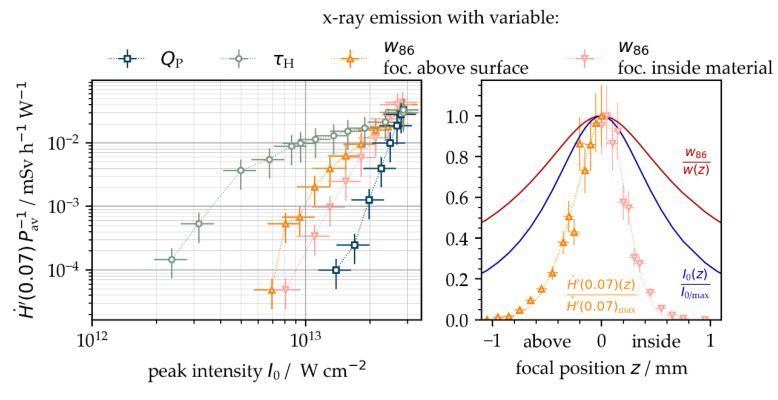
(**Left**): X-ray emission dose rate as function of pulse energy, pulse duration, and laser spot size; (**Right**): effect of laser beam defocusing on laser spot radius, peak intensity, and X-ray emission dose rate obtained at 0.5 MHz PRF, 1.0 µm intra-line pulse distance, and 2.8 m/s ± 0.4/s suction flow speed.

**Table 1 materials-15-02748-t001:** A summary of the influencing factors affecting X-ray photon emissions arising from ultrashort pulse laser materials processing.

Parameter		Impact on X-ray Emission	References
**Laser-sided parameters**	Peak intensity	- Doubling the peak intensity increases the X-ray dose rate by around ten times	[3,12,13,14]
	Pulse energy	- With higher pulse energies, higher X-ray dose rates tend to be expected, - The released X-ray emission increases non-linearly with higher energies, the effective peak intensity has a stronger effect on X-ray emission	[15,16]
	Average laser power	- The released X-ray emission scales approximately linearly with higher average laser powers, pay attention to the non-linear effects with higher pulse energies, - Strongly increased X-ray dose rates for high-PRF laser processes or in burst mode laser processing, in particular when high-PRF pulses irradiate at small geometrical (micro meter) pulse distances	[3,7,16,17]
	Pulse repetition frequency	- X-ray dose rate increases approximately linearly with higher PRF resulting from the accumulation of X-ray emissions respectively released by the individual pulses. - Caution concerning strong increased X-ray emissions for MHz PRF or burst mode laser processing which results from strong laser pulse with plasma interactions	[3,7,16]
	Focus spot diameter	- Doubling the focus spot size at constant peak intensities yields 2.5 times higher X-ray dose rates - The spectral distribution and amplitude shifts towards higher X-ray photon energies with increasing focus spot area	[18]
	Wavelength	- Tendency of lower X-ray dose rates for ultrashort pulses of shorter wavelength	[14]
	Polarisation	- Laser radiation polarized parallel to the scan direction increases the dose rate	[11,16,19]
	Pulse duration	- Tendency of higher X-ray dose rates for ultrashort pulses of longer pulse duration	[15,16,20]
**Process-sided parameters**	Processing regime	- Absorption of the released X-ray emission at the walls of the laser engraved or laser drilled structures reduces the X-ray emission during the ongoing processing - Caution: X-ray dose rate increases as a result of strong laser pulse with plasma/ablation plume interaction, i.e., during stationary laser machining, with deflected beams at high pulse overlap, in laser turning	[7,11,16]
	Scanning direction	- Stronger X-ray emission opposite to the scan direction	[13]
	Intra-line pulse distance	- Pulse irradiations at small geometrical distances cause high surface roughness, in turn lowering the released X-ray dose rate (see surface roughness) - Pulses of small geometrical pulse distances (micro meter) irradiating at MHz PRF amplify the X-ray dose rate by feedback coupling of the pulses with the previously generated laser ablation/plasma plume	[7,11,16]
	Hatch distance	- Larger hatch distances result in higher X-ray dose rates	[11]
	Scan number	- Tendency for lower X-ray dose rates with increasing number of scan crossings	[7]
	Focus position	- Highest X-ray dose rate arises in the focal plane at the position of highest peak intensity - For high-average power laser beams, thermal shift of the optical elements in the beam path can have an effect on the position of highest peak intensity	[2]
	Cross jet	- A larger volume flow rate of the cross jet induces higher X-ray emissions	[16]
	Fume extraction	- A larger distance between sample and suction nozzle decreases the flow rate in turn increasing the X-ray dose rate	[16]
	Angle of incidence	- Oblique laser beams with increased angle of incident with respect to the plasma flank enhance resonance absorption that tends to cause higher X-ray emissions	[7,16,21]
**Material-sided parameters**	Material	- Higher X-ray dose rates occur with elements with a higher atomic number - Highest X-ray dose rate was determined on tungsten, the X-ray dose rate is comparably high on steel and stainless steel materials	[2,12,14]
	Suface roughness	- A higher surface roughness leads to lower X-ray dose rates due to the shielding of the X-rays on microscopic substructures, similar to the shielding effect of boreholes or trenches	[11,16]
	Dimension	- Large-area laser processing under similar irradiation conditions and corresponding X-ray dose rates releases a larger X-ray emission dose	

## Data Availability

Not applicable.

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
