# Peer review of "Enhanced X-ray Emissions Arising from High Pulse Repetition Frequency Ultrashort Pulse Laser Materials Processing"

_materials, 2022, doi:10.3390/ma15082748_

Round 1
Reviewer 1 Report
The authors demonstrated enhanced X-ray emissions from high pulse repetition frequency ultrashort pulse laser materials processing. As shown in table 1, the authors have summarized the influencing factors affecting the X-ray photon emissions arising from ultrashort pulse laser material processing. I think the topic is interesting. The paper is well written, and the text is clear and easy to read. I recommend it to be published in a minor revision.
- The authors should add more discussions on the potential application。
- How about the repeatability of this work?
- Can the other materials observe a similar phenomenon?
Author Response
Dear reviewer,
thank you for reviewing our manuscript. Please find our response to your comments in the file attached.

Reviewer 2 Report
What is the main question addressed by the research?
X-ray emission hazard during laser material processing
Is it relevant and interesting?
Yes
How original is the topic?
Very original
What does it add to the subject area compared with other published material?
X-ray emission hazards during laser ablation were found not too long ago, there is still a very limited amount of papers in this field.
Is the paper well written?
Yes
Is the text clear and easy to read?
Yes
Are the conclusions consistent with the evidence and arguments presented?
Yes
Do they address the main question posed?
Yes
- In equation (2) is the mistake, the square sing is not needed „w^2” on the right-hand side of the equation, please correct.
- The reference [20] is an internet source, I would suggest removing internet type references leaving only scientific paper type references.
- In reference [14] page number and volume are missing, please provide the missing information.
- The whole reference list needs to be carefully reviewed, the formatting needs to be adjusted according to the rules of MDPI.
Author Response

(The authors gave the same response as above.)

Reviewer 3 Report
The authors have attempted to study the enhanced X-ray emissions arising from high pulse repetition frequency ultrashort pulsed lasers and the influence of other laser parameters on X-ray generation.
For example, in AISI 304 stainless steel substrate produce that X-ray photon emission exceeds the legal dose rate limit when ultrashort laser pulses with peak intensities below 1 · 1013 W/cm irradiate at MHz pulse repetition frequency.
It seems work that has been performed well with a novelty of its kind. However, the authors did not project the work that has been done by them in a proper way, since I do believe that there are many gaps in the scripting.
However, the manuscript has to be revised majorly by addressing the queries mentioned in the following, before its publishing in the Materials journal.
- One serious concern is the way it is scripted. It’s not at all creating an impactful and easily one can lose interest. Authors have to focus on this and to improve the readability, the language revision is to be carried out.
- Authors have claimed that X-ray emission is observed to decrease with large beam spot size and longer pulse duration. How do you differentiate the spot size of the laser in ambient air, and on the surface of steel? Authors have to talk about the focusing conditions whether is loosely focussed or tightly focused.
- Please explain, how do you come to a conclusion that they are X-rays, why not gamma rays from the stainless steel.
- How the authors validate, the observed dosage is not due to the background radiation but due to the X-rays generated by the Steel.
- I do believe that the authors did not give a clear explanation for the dependence of generated X-ray dosage on pulse duration and beam waist?
- In the abstract, the unit of peak intensity mentioned is wrong.
- In the abstract, the authors said, they have utilized laser pulses at aMHz, repetition rate. In that case, energy per pulse is of the order nano joules. But authors claim that the max energy per pulse is 36 micro joules, clarify this.
Author Response

(The authors gave the same response as above.)

Round 2
Reviewer 3 Report
Authors have given their versions of the queries and I do believe the manuscript is now acceptable to publish in the MATERIALS.